# Variable-Step Multiscale Fuzzy Dispersion Entropy: A Novel Metric for Signal Analysis

**DOI:** 10.3390/e25070997

**Published:** 2023-06-29

**Authors:** Yuxing Li, Junxian Wu, Shuai Zhang, Bingzhao Tang, Yilan Lou

**Affiliations:** 1School of Automation and Information Engineering, Xi’an University of Technology, Xi’an 710048, China; 2Shaanxi Key Laboratory of Complex System Control and Intelligent Information Processing, Xi’an University of Technology, Xi’an 710048, China

**Keywords:** dispersion entropy, fuzzy dispersion entropy, variable-step multiscale fuzzy dispersion entropy, feature extraction, signal analysis

## Abstract

Fuzzy dispersion entropy (FuzDE) is a newly proposed entropy metric, which combines the superior characteristics of fuzzy entropy (FE) and dispersion entropy (DE) in signal analysis. However, FuzDE only reflects the feature from the original signal, which ignores the hidden information on the time scale. To address this problem, we introduce variable-step multiscale processing in FuzDE and propose variable-step multiscale FuzDE (VSMFuzDE), which realizes the characterization of abundant scale information, and is not limited by the signal length like the traditional multiscale processing. The experimental results for both simulated signals show that VSMFuzDE is more robust, more sensitive to dynamic changes in the chirp signal, and has more separability for noise signals; in addition, the proposed VSMFuzDE displays the best classification performance in both real-world signal experiments compared to the other four entropy metrics, the highest recognition rates of the five gear signals and four ship-radiated noises reached 99.2% and 100%, respectively, which achieves the accurate identification of two different categories of signals.

## 1. Introduction

Entropy, as a nonlinear metric, plays a great role in quantifying the degree of chaos in a system and evaluating the complexity of a time series [1,2,3]. In general, having a high entropy value means that the signal is more complex and difficult to predict, and vice versa [4,5]. In recent years, entropy has been widely employed in biomedical signal analysis, mechanical fault diagnosis, and underwater acoustic signal processing due to its advantages of simple calculation, fast speed, and good robustness [6,7,8].

Along with the continuous deeper research on entropy, new entropy metrics such as sample entropy (SE) [9], fuzzy entropy (FE) [10], permutation entropy (PE) [11], dispersion entropy (DE) [12], etc., have been proposed by numerous scholars successively [13,14,15]. However, these entropies still have limitations. For example, SE reflects the complexity of the signal, but it is complex to calculate, and the entropy value is susceptible to sudden changes in the signal. As an improvement of sample entropy, FE solves the problem of uncertainty of the SE value, but its calculation is still more complicated. PE is simple to calculate but ignores amplitude information. Although the DE considers the amplitude information, its anti-noise performance is poor. To further alleviate the shortcomings of DE, fuzzy dispersion entropy (FuzDE) was proposed [16], which combines the advantages of FE and DE by replacing the circle mapping function of DE with the fuzzy affiliation function in FE to reduce the information loss of the original signal during the calculation.

FuzDE is one of the improved algorithms for DE, and its main improvement lies in the mapping approach of DE. By introducing fuzzy membership degrees, FuzDE overcomes the limitation of traditional mapping approaches that easily lose effective information, so as to improve the detection ability of changes in frequency, amplitude and chaos of time series [17]. However, like other DE-based improvement algorithms, FuzDE also suffers from the defect of single scale, and it is difficult to reflect the effective information of the original series from multiple scales.

To solve the above problems, multiscale versions are derived by combining different coarse-grained methods, which enables these complexity measures to capture information from the temporal scale [18,19,20]. Azami et al. proposed multiscale DE (MDE) and refined composite MDE (RCMDE) in 2017 and applied them to the field of biomedical signal processing, and the research results showed that MDE and RCMDE can effectively respond to signal information on different time scales with shorter computation time and can better distinguish different types of physiological signals compared with the previous entropy metrics [21]. In 2023, Li proposed a multiscale FuzDE (MFuzDE) and applied it to the domain of hydroacoustic signal processing, and the results of empirical experiments showed that MFDE was better than MDE for ship radiation noise analysis [22].

Although the above metrics alleviated the problem that FuzDE could not respond to signal information at multiple scales to a certain extent, these metrics still suffered from the loss of signal subsequence information with the increase in time series scale, and there were certain computational errors. In 2022, variable-step multiscale Lempel-Ziv complexity was proposed, which retains more potential information through the feature of variable step processing process and solves the problem of sequence length shortening with increasing scale [23]. Inspired by the advantages of variable-step multiscale processing, we proposed variable-step multiscale FuzDE (VSMFuzDE) by incorporating it into FuzDE.

In general, the main contributions of this paper are as follows: based on FuzDE, the VSMFuzDE is proposed by introducing variable-step multiscale processing; compared with FuzDE, VSMFuzDE can obtain rich scale information and can provide more stable performance for signal analysis when the scale factor (SF) is large. The structure of the rest of the paper is as follows: Section 2 describes FuzDE and the proposed VSMFuzDE in detail; Section 3 proves the validity of VSMFuzDE using different simulated signals; Section 4 conducts experiments on the extraction and classification of two classes of real-world signals based on VSMFuzDE; the main conclusions of the entire paper are shown in Section 5.

## 2. Fundamental Theory

### 2.1. Fuzzy Dispersion Entropy

The fuzzy dispersion entropy (FuzDE) introduces the fuzzy membership function on the basis of DE as a substitute of the round function of DE, which further improves the noise immunity and stability of DE. The calculation steps of FuzDE are as follows:

Step 1: For specific time series $X=\left\{x_{1},x_{2},\ldots ,x_{N}\right\}$, the series X is transformed to a new series $Y=\left\{y_{1},y_{2},\ldots ,y_{N}\right\}$ by normal cumulative distribution function (NCDF), and the NCDF is described as follows:(1)$$y_{i}=\frac{1}{\sigma \sqrt{2\pi }}\int_{-\infty }^{x_{i}}e^{-\frac{{\left(t-\mu \right)}^{2}}{2\sigma^{2}}}dt\;\left(i=1,2,\ldots ,N\right)$$
where σ and μ are the standard deviation and mean value of X respectively.

Step 2: The series Y is mapped to a new symbolic sequence
$Z^{c}=\{z_{1}^{c},z_{2}^{c},\ldots ,z_{N}^{c}\}$ according to the Formula (2):(2)$$z_{i}^{c}=cy_{i}+0.5$$
where c is the number of categories.

Step 3: Introduce the fuzzy membership function on the sequence $Z^{c}$ as follows:(3)$$\mu_{M_{1}}\left(z_{i}^{c}\right)=\left\{\begin{array}{c} 0\;\;\;\;\;\;\;\;\;\;\;\;\;\;\;\;\;\;\;\;z_{i}^{c}\geq 2\\ 2-z_{i}^{c}\;\;\;\;\;\;\;\;\;\;\;\;\;\;1\leq z_{i}^{c}<2\;\;\;\;\\ \;\;\;\;\;\;1\;\;\;\;\;\;\;\;\;\;\;\;\;\;\;0\leq z_{i}^{c}<1\; \end{array}\right.$$
(4)$$\mu_{M_{k}}\left(z_{i}^{c}\right)=\left\{\begin{array}{c} 0\;\;\;\;\;\;\;\;\;\;\;\;\;z_{i}^{c}\leq k-1\\ z_{i}^{c}-k+1\;\;\;\;\;k-1<z_{i}^{c}\leq k\\ k+1-z_{i}^{c}\;\;\;\;\;\;k<z_{i}^{c}<k+1\\ 0\;\;\;\;\;\;\;\;\;\;\;\;\;z_{i}^{c}\geq k+1 \end{array}\right.$$
(5)$$\mu_{M_{c}}\left(z_{i}^{c}\right)=\left\{\begin{array}{c} 0\;\;\;\;\;\;\;\;\;\;\;\;\;\;\;\;\;\;\;\;\;\;z_{i}^{c}\leq c-1\\ 1+c-z_{i}^{c}\;\;\;\;\;\;\;\;\;\;\;\;c-1<z_{i}^{c}<c\;\\ 1\;\;\;\;\;\;\;\;\;\;\;\;\;\;\;\;\;\;\;\;\;\;\;\;\;\;\;\;\;z_{i}^{c}\geq c \end{array}\right.$$
where $M_{k}$ is the fuzzy membership function, and k stands for the kth class. $\mu_{M_{k}}\left(z_{i}^{c}\right)$ is the degree of membership of $z_{i}^{c}$ for the kth class. Through the fuzzy membership function, each $z_{i}^{c}$ will have one or two different degrees that are integers between $\left[1,\;c\right]$.

Step 4: The sequence $Z^{c}$ of Step 3 is reconstructed into $N-\left(m+1\right)\tau$ subsequences $Z_{j}^{m,c}$ by phase space reconstruction as follows:(6)$$Z_{j}^{c,m}=\left\{z_{j}^{c},z_{j+\tau }^{c},\ldots ,z_{j+\left(m-1\right)\tau }^{c}\right\},\;\;j=1,2,\ldots ,N-\left(m-1\right)\tau$$
where m is the embedding dimension, and τ is the delay time.

Step 5: Each vector $Z_{j}^{c,m}$ is assigned to the dispersion patterns $\pi_{v_{0}v_{1}\ldots v_{m-1}}$, where $v_{0}$ is $z_{j}^{c}$, $v_{1}$ is $z_{j+\left(1\right)\tau }^{c}$, and $v_{m-1}$ is $z_{j+\left(m-1\right)\tau }^{c}$. Then, the membership degree of each vector $Z_{j}^{c,m}$ is calculated to obtain the membership degree of each dispersion pattern:(7)$$\mu_{\pi_{v_{0}v_{1}\ldots v_{m-1}}}\left(Z_{j}^{c,m}\right)=\prod_{i=0}^{m-1}\mu_{M_{v_{i}}}\left(z_{j+\left(i\right)\tau }^{c}\right)$$
in general, the number of dispersion patterns that attributed to each vector $Z_{j}^{c,m}$ in FuzDE is $c^{m}$, which is similar to DE.

Step 6: The probability of each dispersion pattern is computed according to Equation (8), and the equation is described as follows:(8)$$p\left(\pi_{v_{0}v_{1}\ldots v_{m-1}}\right)=\frac{\sum_{j=1}^{N-\left(m-1\right)d}\mu_{\pi_{v_{0}v_{1}\ldots v_{m-1}}}\left(Z_{j}^{m,c}\right)}{N-\left(m-1\right)\tau }$$

Step 7: The FuzDE can be calculated according to the theory of Shannon entropy as follows:(9)$$\mathrm{FuzDE}\left(X,m,c,\tau \right)=-\overset{c^{m}}{\underset{\pi =1}{\sum }}p\left(\pi_{v_{0}v_{1}\ldots v_{m-1}}\right)lnp\left(\pi_{v_{0}v_{1}\ldots v_{m-1}}\right)$$

Step 8: The normalized FuzDE (NFuzDE) is defined as follows:(10)$$\mathrm{NFuzDE}\left(X,m,c,\tau \right)=\frac{FuzDE\left(X,m,c,\tau \right)}{ln\left(c^{m}\right)}$$

### 2.2. Variable-Step Multiscale Fuzzy Dispersion Entropy

FuzDE improves the stability and noise immunity of DE, but still has the same drawback as DE, that it cannot explore the abundant information on the time scale. For this reason, we adopted variable-step multiscale processing, which is free from the limitation of traditional multiscale and refine composite multiscale processing depending on sequence length, and combined it with FuzDE to propose VSMFuzDE, the specific steps are as follows:

Step 1: The specific time series $X=\left\{x_{1},x_{2},\ldots ,x_{N}\right\}$ is converted into s subsequences by the following variable-step multiscale analysis:(11)$$y_{d,l}^{\left(S\right)}=\frac{1}{s}\overset{d\left(l-1\right)+S}{\underset{i=d\left(l-1\right)+1}{\sum }}x_{i},1\leq d\leq s,\;1\leq l\leq \frac{N-S}{d}+1$$
where S is the scale factor (SF), d is the moving step length, and $y_{d,l}^{\left(S\right)}$ is the l-th element when the moving step length is d. It can be found that the subsequence is the same as the original sequence when S=1.

Step 2: The NFuzDE value for the subsequence with step length d is calculated by the as following formula:(12)$$\mathrm{NFuzDE}\left(y_{d}^{\left(S\right)},m,c,\tau \right)=\frac{\mathrm{FuzDE}\left(y_{d}^{\left(S\right)},m,c,\tau \right)}{ln\left(c^{m}\right)}$$

Step 3: The NFuzDE values of all subsequences obtained in Step 1 are calculated and the mean value of NFuzDE of all subsequences is defined as the VSMFuzDE value:(13)$$\mathrm{VSMFuzDE}\left(X,m,c,\tau ,S\right)=\frac{1}{S}\overset{S}{\underset{d=1}{\sum }}\mathrm{NFuzDE}\left(y_{d}^{\left(S\right)},m,c,\tau \right)$$

For a more intuitive understanding of the variable-step coarse-grained analysis, the coarse-grained process for a scale factor of three is drawn. Figure 1 shows the variable-step multiscale analysis for scale factor S=3.

## 3. Validation of Simulation Signal

In this section, the superior performance in signal analysis of the proposed VSMFuzDE is demonstrated by conducting two sets of synthetic signal experiments. Among them, the synthetic signals include noise signals and chirp signals, and the entropies used for comparison contain RCMFuzDE, MFuzDE, RCMDE, and MDE. To ensure the reliability of the experiments, these parameters are set consistently, the embedding dimension m and time delay τ are three and one, respectively, and the number of categories c is uniformly set to six [24].

### 3.1. Validation of Separability

In this section, a classical noises separation experiment is performed with reference to [25]. Overall, 50 independent white noises and pink noises are taken as 50 samples, and the sample length is 2048, and then the different entropy values of these noises on a scale of 1 to 10 are calculated, respectively. The mean and standard deviation of the entropy values are obtained by calculating them as shown in Figure 2. In addition, the experiment uses ANOVA on entropy mean curves with signal type as an independent factor to assess the separability of the five entropies for white noise and pink noise. The *p*-values obtained by ANOVA for different entropy are shown in Table 1.

Figure 2 and Table 1 show that the mean entropy curves of MDE and RCMDE are very close to each other when scale factor is 4, and the *p*-values obtained by ANOVA are greater than 0.05, which indicates that there is no significant difference between white noise and pink noise; for MFuzDE, RCMFuzDE and the proposed VSMFuzDE, the mean curves are obviously different, and there is a significant difference between the two noise signals as the *p*-values are less than the confidence level of 0.05; in addition, compared with MFuzDE and RCMFuzDE, the proposed VSMFuzDE has the smaller standard deviation of the two noises at each scale factor and no overlapping parts, which indicates that VSMFuzDE has stronger stability. In summary, the proposed VSMFuzDE has better separability and stronger stability compared to the other four entropies.

### 3.2. Validation of the Ability to Detect Dynamic Changes

The ability to detect dynamic changes is an important property of nonlinear dynamic metrics. In this experiment, the property of the proposed VSMFuzDE is verified by choosing the chirp signal. And the chirp signal is defined as:(14)$$x\left(t\right)=exp\left(j2\pi \left(f_{0}t+\frac{1}{2}kt^{2}\right)\right)$$
where $f_{0}$ is the initial frequency, taken as 10 Hz, and k is the modulation frequency, taken as 3. The frequency of the chirp signal was set from 10 Hz to 100 Hz and made to last for 20 s at a sampling frequency of 1000 Hz; thus, 20,000 sampling points can be obtained.

The length of the sliding window is taken as 1000 sampling samples, and then the sliding sampling is performed at 90% overlap rate, thus the 20,000 sampling points are divided into a total of 190 samples by sliding sampling. The VSMFuzDE value of each sample was calculated. The chirp signals and the corresponding different entropy curves are shown in Figure 3 and Figure 4, respectively.

According to Figure 4, it can be seen that the curves of the multiscale version of DE fluctuate greatly under all scale factors, the different multiscale versions of FuzDE can reflect the change of signal complexity, and the dynamic change detection ability is better than the various multiscale versions of DE; among them, when S=5, the curves of RCMFuzDE and MFuzDE appear in a downwards trend, which is not in consistent with the signal complexity growth trend, only the proposed VSMFuzDE has the most stable trend, and the corresponding curve is smoother. Therefore, it can be inferred from the experiment that VSMFuzDE is more sensitive to changes in signal complexity than other entropies.

## 4. Validation of the Realistic Signal

To demonstrate the effectiveness of the proposed VSMFuzDE in signal analysis, feature extraction experiments are conducted for the gear signal and ship-radiated noise, respectively, in this section. Moreover, the complexity metrics used for comparison and the corresponding parameter settings are consistent with those in the simulation experiments.

### 4.1. Validation of Gear Signal

In this section, five gear signals from the Southeastern University [26] are chosen, which are Health working state, Missing tooth, Chipped tooth, Root fault, and Surface fault. For convenience, these five gear signals are named as Health, Missing, Chipped, Root, and Surface. The time domain waveform of the gear signal is demonstrated in Figure 5.

For each state of the gear signal, 409,600 sampling points are intercepted and equally divided into 200 samples without overlap, with each sample containing 2048 sampling points.

Then, different entropy values of each signal are calculated as its features, the scale factor is taken as 10, so each signal will obtain a feature set with dimension (200×10); to facilitate comparison, 50% of the samples are randomly selected as the training set and the rest as the test set, and obtain the average recognition rates of the five gear signals at each scale by the K-nearest neighbor algorithm (KNN) [27]. Figure 6 shows the entropy values and standard deviations of five entropy indicators for gear signals at different scales. The average recognition rate of the five gear signals at each scale are presented in Table 2.

As can be seen from Figure 6, the interval of the feature value curves of the signals is obvious only in the plot of VSMFuzDE, and the feature value curves of the signals of the remaining metrics are almost mingled together; as the scale increases the differentiation performance of VSMFuzDE on the signal increases significantly, and the remaining four indicators do not change obviously on the differentiation performance of the signal. Therefore, the following conclusion can be obtained: VSMFuzDE has the best differentiation effect on the five gear signals.

From Table 2, it can be seen that the proposed VSMFuzDE has the highest average recognition rate at each scale; except for VSMFuzDE, the average recognition rate of entropy decreases significantly when the scale factor is large, and the lowest recognition rate occurs at S=10; in addition, only VSMFuzDE maintains excellent recognition performance when the scale factor is large, and reaches a maximum of 82.2% at a scale factor of 8. As a result, the proposed VSMFuzDE has the best outstanding performance in feature extraction at each scale.

Although the above results reveal that VSMFuzDE has better classification performance at different scales, the above classification experiments are discussed only from a single feature and the recognition rate of five gear signals is not desirable. For this reason, this experiment combines different scales for multifeature classification and recognition. Here, (1, 7) to represent the combination of features with scale factors of 1 and 7, (1, 3, 8) to represent the combination of features with scale factors of 1, 3 and 8, and so on. The highest recognition rates of gear signals under different numbers of extracted features are shown in Table 3.

Table 3 indicates that the recognition rate of each entropy increases with the number of scales; among them, the recognition rate of MDE is the lowest with only 84.2% for different numbers of features, which shows that the MDE has the worst ability to distinguish the five gear states; and the proposed VSMFuzDE has the highest recognition rate and reaches a high of 99.2% when the number of features is three, which is much higher than RCMFuzDE and MFUzDE. This experiment shows that the combination of multiple features for VSMFuzDE improves the separability of gear signals and further confirms the effectiveness of the proposed VSMFuzDE in analyzing complex signals.

To intuitively express the effect of feature extraction with different entropies, the obtained features of 10 scales are reduced to two dimensions by t-stochastic neighbor embedding (t-SNE), and then the visualization of the features is carried out. Figure 7 shows the visualization of the features for gear signals by t-SNE.

From Figure 7, it can be concluded that the Missing fault state has a large overlap area with the other four states for RCMFuzDE, MFuzDE, RCMDE and MDE, making it difficult to distinguish; among them, MDE has the largest overlap area and the worst classification effect; in addition, only the proposed VSMFuzDE has the best effect on the identification of the five fault states, having the least overlapping samples. In general, the visualization results of VSMFuzDE show the smallest overlap and the best clustering, which indicates that VSMFuzDE has a stronger ability to discriminate five types of gear signals.

### 4.2. Validation of Ship-Radiated Noise

In this section, the data from the national parks [27] were adopted, and they have been named Ship-1, Ship-2, Ship-3, and Ship-4. The length of each signal and the number and length of samples are the same as in Section 4.1. The time domain waveform of the ship-radiated noise signal is presented in Figure 8.

To reflect the degree of differentiation of these metrics more accurately on the signal, KNN was employed for classification experimental knowledge. The entropy values and standard deviations of five entropy metrics for ship-radiated noise signals at different scales are shown in Figure 9. Table 4 illustrates the average recognition rate of the four ship-radiated noise signals at each scale, and Table 5 shows the highest recognition rates of the ship-radiated noise signal under different numbers of extracted features.

It can be seen from Figure 9, that in general, the entropy curves of all signals first rise with increasing scale and then stabilize; with the increase in scale, only the distance of the entropy curves of the signals in the VSMFuzDE figure increases, and the entropy curves in the rest of the figures decrease, among which the decreasing trend of MDE and DE is the most obvious; Ship-2 is the most difficult to distinguish in all subplots with the rest of the radiated noise signals having overlapping parts. In general, VSMFuzDE has the best separation of the four ship radiation noise signals compared to the other four entropy metrics.

As can be seen from Table 4, the recognition rate of most of the indicators for the signals under 10 scales appeared to be less than 65%, and only the recognition rate of VSMFuzDE signals was higher than 65% under each scale; VSMFuzDE had the highest recognition rate for the signals under scale factor 8 and reached 81.5%, RCMFuzDE and MFuzDE had the highest recognition rate for the signals under only one scale. It can be concluded that VSMFuzDE has a better performance in distinguishing signals compared to other metrics.

As shown in Table 4 and Table 5, compared with the recognition rate in Table 4, the recognition rates of all metrics in Table 5 have improved dramatically; only the recognition rate of VSMFuzDE for signals reaches 99.5% above the recognition rate under different numbers of features and reaches 100% under the combination of three feature numbers. In short, compared with other metrics, VSMFuzDE has the highest recognition rate for signals and better classification performance.

This experiment is the same as the gear signal experiment, representing the distribution of signal characteristics more intuitively. The features obtained at 10 scales by t-SNE visualization; Figure 10 displays the visualization of the ship-radiated noise features by t-SNE.

From Figure 10, Ship-2 is the most independent among all sub-figures, and Ship-1 always overlaps with other signals; the Ship-1 and Ship-3 signals in RCMFuzDE and MFuzDE, respectively, appear to have chunking into two blocks; the signal in VSMFuzDE has the densest feature distribution with the least overlapping samples; moreover, the aggregation performance of the samples of Ship-4 of RCMDE is not strong, and the samples of Ship-2, Ship-3, and Ship-4 of MDE have overlapping phenomena. Therefore, it can be proven that VSMFuzDE has a better ability to distinguish the ship-radiated noise than other metrics.

## 5. Conclusions

A new metric, called VSMFuzDE, was developed in this paper, which enhances the ability of FuzDE to analyze signals from a coarse-grained processing perspective; moreover, VSMFuzDE is validated by simulated signals and applied to feature extraction real-word signals. The main conclusions can be generalized as follows:

(1) The VSMFuzDE is proposed by combining the variable-step multiscale process, which not only solves the problem that the FuzDE cannot respond to the multiscale information of the signal but also improves the problem of information loss caused by the increase in scale factor in the calculation process of the traditional multiscale method.

(2) The experimental simulation results show that VSMFuzDE has significantly better separability for noise signals than other metrics and has the most sensitive dynamic change detection for chirp signals. The effectiveness of VSMFuzDE in the signal analysis is verified by these studies.

(3) The real-world signal experiments show that in the same number of features compared with other metrics, VSMFuzDE has the best ability to distinguish the signal. In particular, when identifying gear signals, the recognition rate of the feature extraction method based on VSMFuzDE reaches 99.5% when taking three features, which is at least 11% higher than the other methods.

## Figures and Tables

**Figure 1 entropy-25-00997-f001:**
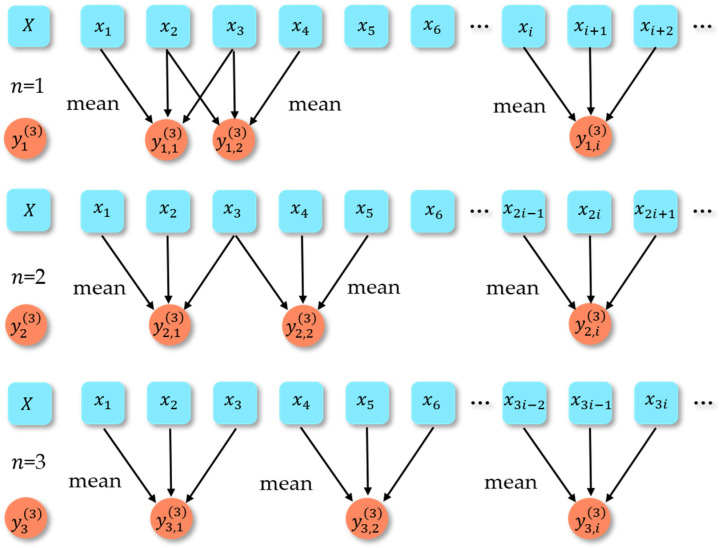
The variable-step multiscale processing for scale factor S=3.

**Figure 2 entropy-25-00997-f002:**
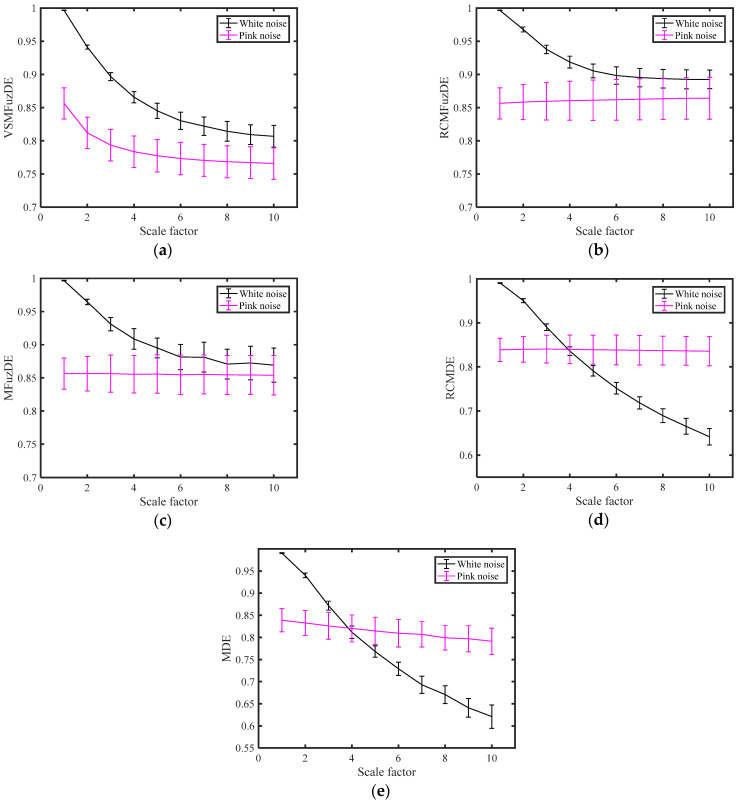
The mean and standard deviation of the entropy values for white noise and pink noise. (**a**) VSMFuzDE, (**b**) RCMFuzDE, (**c**) MFuzDE, (**d**) RCMDE, (**e**) MDE.

**Figure 3 entropy-25-00997-f003:**
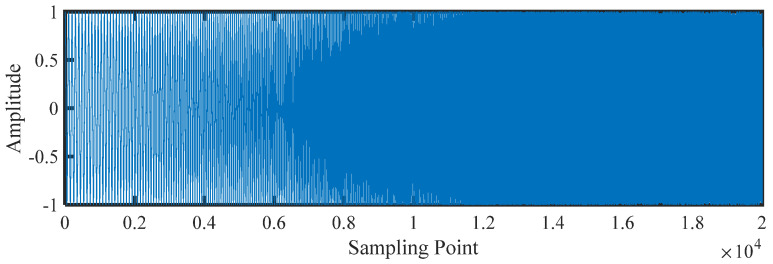
The chirp signals.

**Figure 4 entropy-25-00997-f004:**
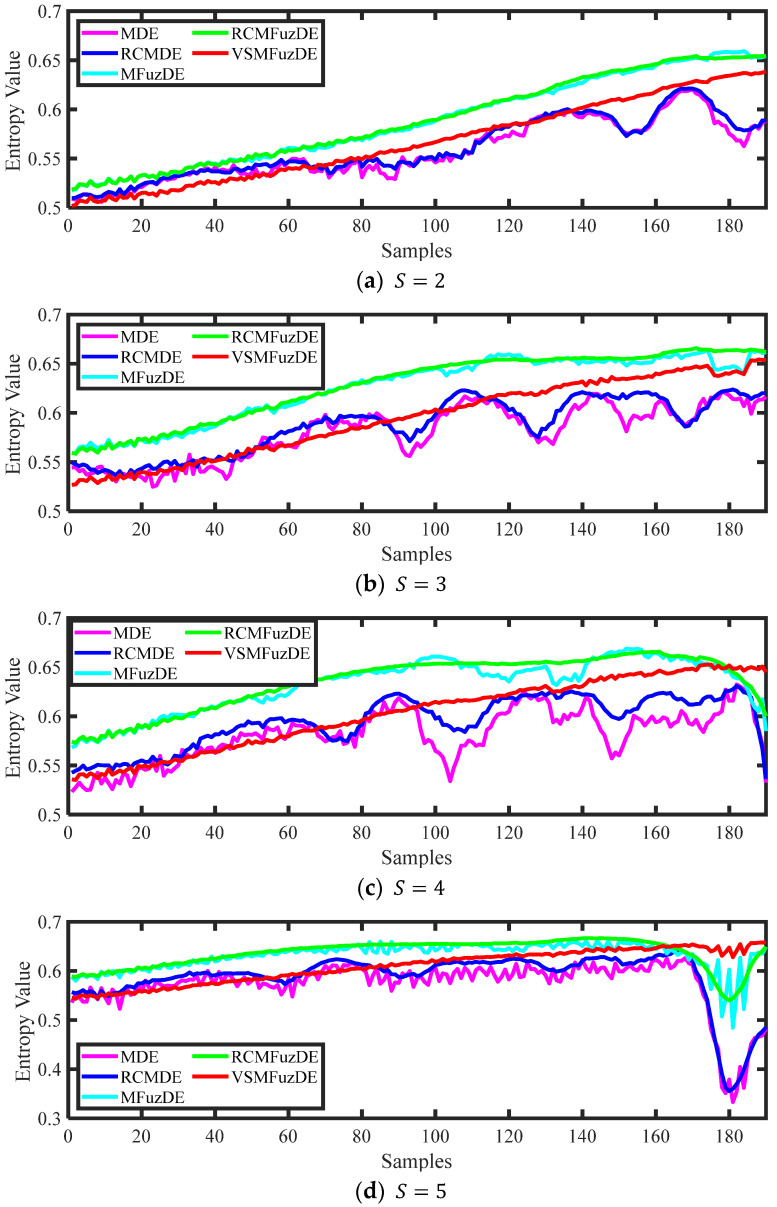
The different entropy curves of chirp signals.

**Figure 5 entropy-25-00997-f005:**
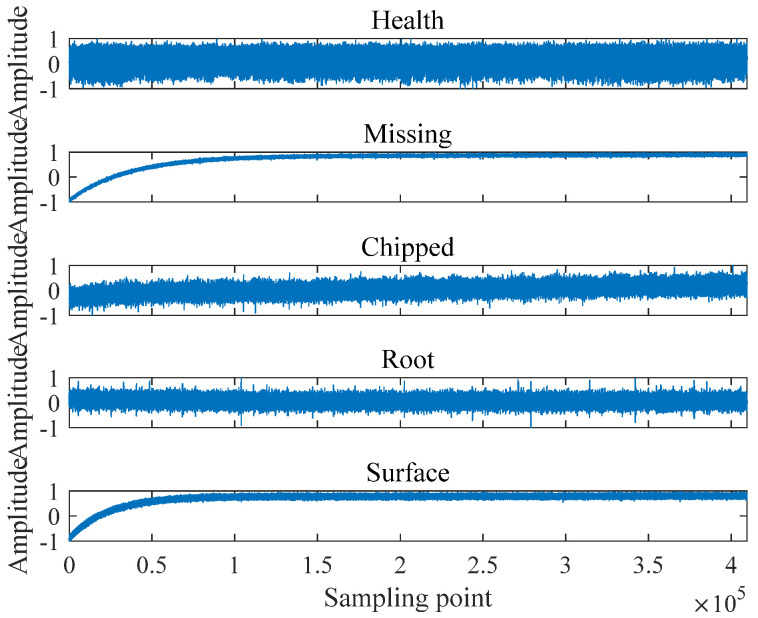
The time domain waveform of the gear signal.

**Figure 6 entropy-25-00997-f006:**
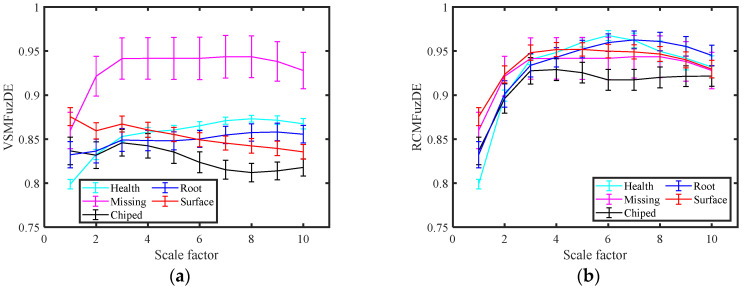
The entropy values and standard deviations of five entropy metrics for gear signals at different scales. (**a**) VSMFuzDE, (**b**) RCMFuzDE, (**c**) MFuzDE, (**d**) RCMDE, (**e**) MDE.

**Figure 7 entropy-25-00997-f007:**
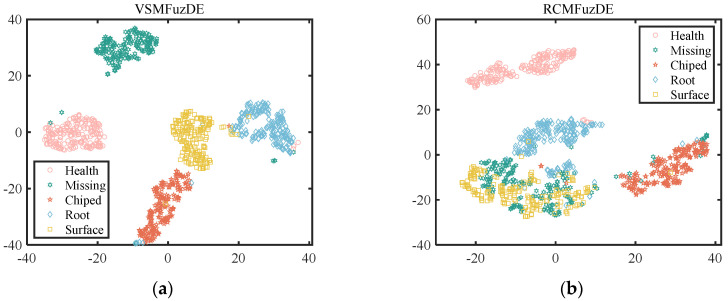
Visualization of the features for gear signals by t-SNE. (**a**) VSMFuzDE, (**b**) RCMFuzDE, (**c**) MFuzDE, (**d**) RCMDE, (**e**) MDE.

**Figure 8 entropy-25-00997-f008:**
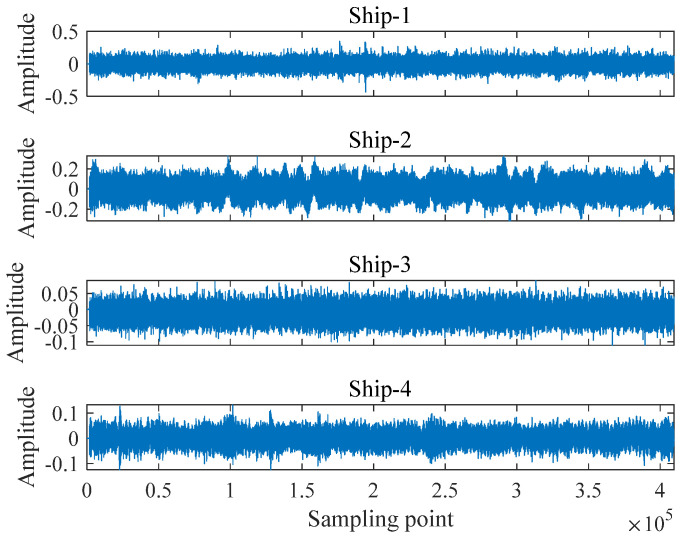
The time domain waveform of the ship-radiated noise signal.

**Figure 9 entropy-25-00997-f009:**
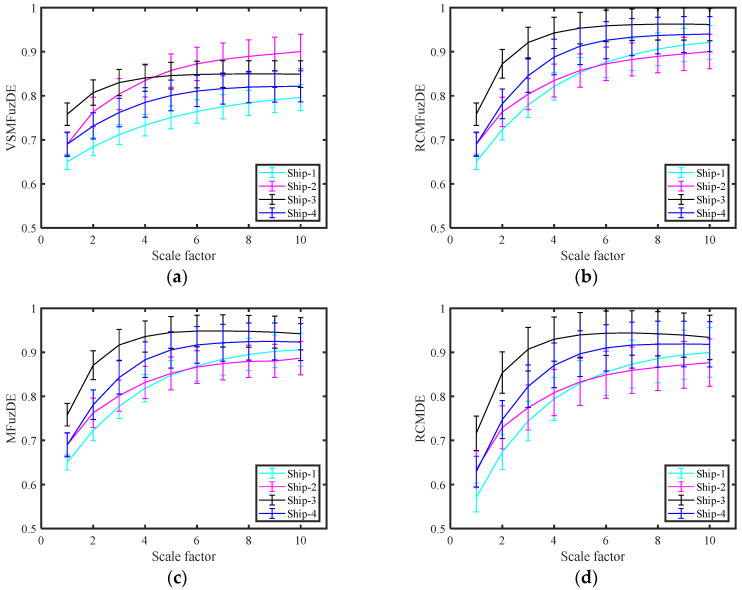
The entropy values and standard deviations of five entropy metrics for ship-radiated noise signals at different scales. (**a**) VSMFuzDE, (**b**) RCMFuzDE, (**c**) MFuzDE, (**d**) RCMDE, (**e**) MDE.

**Figure 10 entropy-25-00997-f010:**
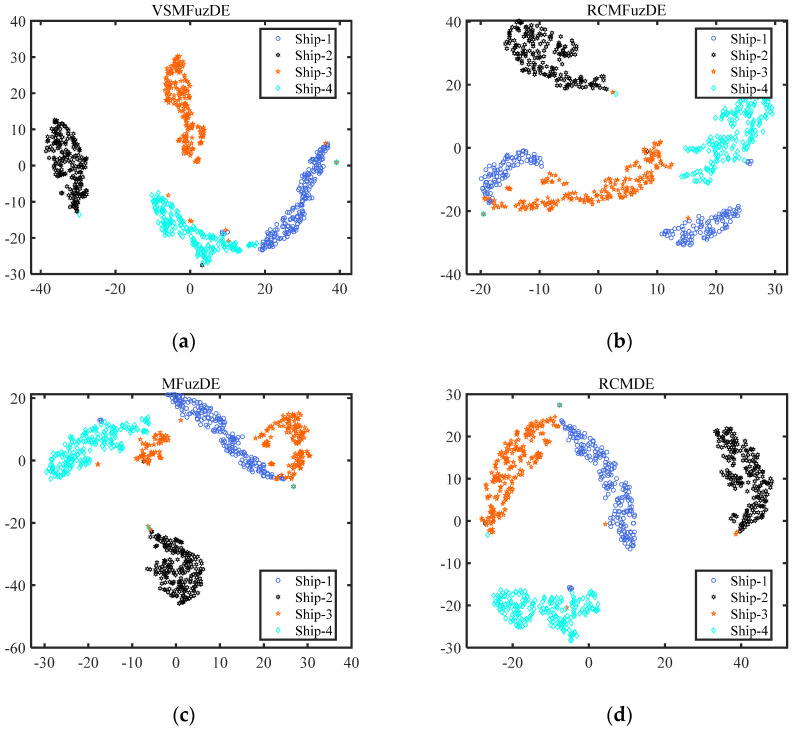
The visualization of the ship-radiated noise features by t-SNE. (**a**) VSMFuzDE, (**b**) RCMFuzDE, (**c**) MFuzDE, (**d**) RCMDE, (**e**) MDE.

**Table 1 entropy-25-00997-t001:** The *p*-values obtained by ANOVA for different entropy.

Metric	*p*-Value
VSMFuzDE	0.0029
RCMFuzDE	7.853$\;\times \;10^{-5}$
MFuzDE	0.0015
RCMDE	0.2459
MDE	0.3376

**Table 2 entropy-25-00997-t002:** The average recognition rate of the five gear signals at each scale (%).

Metric	Scale Factor									
	1	2	3	4	5	6	7	8	9	10
VSMFuzDE	57.6	53.4	53.4	48.0	54.8	68.4	79.0	82.2	75.8	70.0
RCMFuzDE	57.6	33.0	32.0	34.6	37.4	46.0	45.0	39.6	34.4	25.4
MFuzDE	57.6	32.8	30.0	26.8	35.6	47.6	41.0	30.8	27.8	25.0
RCMDE	53.8	32.6	31.8	30.0	34.0	47.2	41.8	33.4	28.2	26.0
MDE	53.8	34.2	26.4	25.4	34.2	39.0	33.0	27.0	26.8	21.6

**Table 3 entropy-25-00997-t003:** The highest recognition rates of gear signals under different numbers of extracted features (%).

Metric	ARR/SF	Number of Extracted Features			
		2	3	4	5
VSMFuzDE	ARR	97.4	99.2	99.2	99.2
	SF	(1, 7)	(1, 3, 8)	(1, 2, 3, 8)	(1, 2, 3, 4, 8)
RCMFuzDE	ARR	77.0	88.2	91.8	92.4
	SF	(1, 6)	(1, 2, 7)	(1, 2, 3, 6)	(1, 2, 3, 5, 6)
MFuzDE	ARR	77.0	86.4	88.6	88.8
	SF	(1, 7)	(1, 2, 6)	(1, 2, 3, 7)	(1, 2, 3, 4, 7)
RCMDE	ARR	79.8	85.6	89.8	90.4
	SF	(1, 7)	(1, 2, 7)	(1, 2, 4, 7)	(1, 2, 4, 5, 7)
MDE	ARR	73.8	79.0	80.0	84.2
	SF	(1, 7)	(1, 2, 6)	(1, 2, 5, 7)	(1, 2, 5, 7, 9)

**Table 4 entropy-25-00997-t004:** The average recognition rate of the four ship-radiated noise signals at each scale.

Metric	Scale Factor									
	1	2	3	4	5	6	7	8	9	10
VSMFuzDE	72.25	76.5	74.75	66.00	76.50	78.5	79.50	78.25	81.5	78.25
RCMFuzDE	72.25	73.5	70.25	70.00	68.25	66.25	64.25	63.25	57.75	51.25
MFuzDE	72.25	70.25	68.00	67.00	65.00	64.50	59.50	58.25	55.25	52.75
RCMDE	75.75	72.00	74.25	65.50	66.50	62.75	60.50	61.75	48.75	46.25
MDE	75.25	70.50	68.75	69.00	64.75	62.25	55.25	44.75	45.75	39.50

**Table 5 entropy-25-00997-t005:** The highest recognition rates of ship-radiated noise signals under different numbers of extracted features (%).

Metric	ARR/SF	Number of Extracted Features			
		2	3	4	5
VSMFuzDE	ARR	99.75	100	99.75	99.75
	SF	(1, 3)	(1, 2, 3)	(1, 2, 3, 5)	(1, 2, 3, 4, 5)
RCMFuzDE	ARR	99.00	99.75	99.75	99.75
	SF	(1, 5)	(1, 3, 6)	(1, 2, 5, 7)	(1, 2, 3, 4, 6)
MFuzDE	ARR	98.75	99.75	99.75	99.75
	SF	(1, 3)	(1, 4, 8)	(1, 2, 3, 8)	(1, 2, 3, 5, 8)
RCMDE	ARR	99.25	99.50	99.50	99.25
	SF	(1, 3)	(1, 2, 3)	(1, 3, 9, 10)	(1, 2, 3, 4, 9)
MDE	ARR	99.25	99.50	99.75	99.75
	SF	(1, 3)	(1, 3, 8)	(1, 4, 8, 9)	(1, 2, 4, 8, 9)

## Data Availability

The data used to support the findings of this study are available from the corresponding author upon request.

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
