# Peer review of "Variable-Step Multiscale Fuzzy Dispersion Entropy: A Novel Metric for Signal Analysis"

_entropy, 2023, doi:10.3390/e25070997_

Round 1

Reviewer 1 Report

In this manuscript, the authors proposed a method for signal analysis using variable-step multiscale fuzzy dispersion. The experiments verified the effectiveness of the proposed method, and the results are promising. Overall, I believe that the manuscript requires minor revisions. The following comments are intended to help the authors improve their work:
(1) What is the rationale behind determining the parameters for each respective method?
(2) In Section4, it is necessary for the authors to generate entropy curves for various signals in order to visually illustrate the impact of different entropy methods.
(3) The authors should employ consistent eigen value scales for comparative analysis in Tables2 and 4. Moreover, what is the justification for selecting different eigen values?
(4) The font size of certain images is insufficient and should be adjusted accordingly.
Overall, I believe this article has high academic value but requires some further refinement. I look forward to see the authors' improvements in the manuscript and to reading a more polished version in the near future.

Author Response

Please see attachment for response details.

Reviewer 2 Report

L47 -- "tend to lose" please revise

L.157 -- "we set", please avoid using peronal structures and active voice. 

L.161 -- "we only give" please avoid use of active voice, a more formal language is encourged to be used. 

Section 3.1, Displaying the results in figures in quite intuitive but it would rather be supported by statistical inference or statistical tests, i.e ANOVA, Kruskall-wallis, mann-whitney or Ttest 2

Section 3.2, Why only scales 2 and 5 are given? is there any reason for choosing this scales? 

Please explain better the experimental setup regarding the segments. Its quite confussing, what relation exists between Sample Point in chirp signal and samples in complexity value?

Figure 4: what do the sample in X axis refers to? signal length? 

Similarly, giving results is figures is illustative, however it would be better if a regression and R2 coefficient are diplayed, supporting the above results.

Section 4.1: Why no results for complexity values are given? how can it be seen that the behaviour over these signals adjusts to the prior stated behaviour in section 3? Please give proof 

What is the point on recognition? beeing able to identify between which of the 5 signals is being analyzed? Please explain what was the experiment conducted for results given in Table 1, not just the obtained results. 

Well the term scale factor is confussing regarding features. Features, usually refers to different measures, or characteristics of a signal, ie. mean/median/std/cx value.. No different scales. Please clarify if each feature corresponds to a scale factor or it means something different. 

Figure 6, what do the axis represent in these figures. It is complexity fvalues? i guess not as it has negative values. So what are these representing? Please indicate. 

Section 4.2: please provide results for complexity values 

Figure 7: similar comments for figure 6. 

Statistical analysis supporting figure results are needed. 

A general conclusion is needed. 

References: please attend formatting, some journals appear in italics while other do not. Webpages should provide date of access to resource. Note identation in reference 20 

Too much use of structures "we... " a more formal english language is needed. 

Author Response

(The authors gave the same response as above.)

Round 2

Reviewer 1 Report

The author has made detailed revisions to the manuscript, and the reviewer suggests accepting this paper.

Minor editing of English language required.

Author Response

Thank you very much for your work and consideration in reviewing our paper. On behalf of my co-authors, I would like to express my sincere gratitude to you.

Reviewer 2 Report

Find attach file. 

Still some changes to be done. 

Author Response

Thank you for your review, we have replied to your comments. Please check the attachment for details

Round 3

Reviewer 2 Report

Still think statistical analysis is needed in order to support graphical results, but the improvements are notorious. Only would suggest preserving axis limits equal in all subfigures in each figure. 

Author Response

Thank you for your comments on our manuscript. We have revised the manuscript based on your suggestions, and the specific changes are in the attachment.
